# One-Part Alkali-Activated Materials: State of the Art and Perspectives

**DOI:** 10.3390/polym14225046

**Published:** 2022-11-21

**Authors:** Yongjun Qin, Changwei Qu, Cailong Ma, Lina Zhou

**Affiliations:** 1College of Civil Engineering and Architecture, Xinjiang University, Urumqi 830047, China; 2Xinjiang Civil Engineering Technology Research Center, Urumqi 830017, China; 3Xin Jiang Key Lab of Building Structure and Earthquake Resistance, Xinjiang University, Urumqi 830047, China; 4College of Mathematics and System Sciences, Xinjiang University, Urumqi 830047, China

**Keywords:** alkali-activated, one-part, geopolymer, mix design, mechanical properties, durability

## Abstract

Alkali-activated materials (AAM) are recognized as potential alternatives to ordinary Portland cement (OPC) to limit CO_2_ emissions and beneficiate several wastes into useful products. Compared with its counterparts involving the concentrated aqueous alkali solutions, the development of “just add water” one-part alkali-activated materials (OP-AAM) has drawn much attention, mainly attributed to their benefits in overcoming the hazardous, irritating, and corrosive nature of activator solutions. This study starts with a comprehensive overview of the OP-AAM; 89 published studies reported on mortar or concrete with OP-AAM were collected and concluded in this paper. Comprehensive comparisons and discussions were conducted on raw materials, preparation, working performance, mechanical properties, and durability, and so on. Moreover, an in-depth comparison of different material pretreatment methods, fiber types, and curing methods was presented, and their potential mechanisms were discussed. It is found that ground granulated blast-furnace slag (GGBS) provides the best mechanical properties, and the reuse of most aluminosilicate materials can improve the utilization efficiency of solid waste. The curing temperature can be improved significantly for precursor materials with low calcium contents. In order to overcome the brittleness of the AAM, fiber reinforcement might be an efficient way, and steel fiber has the best chemical stability. It is not recommended to use synthetic fiber with poor chemical stability. Based on the analysis of current limitations, both the recommendations and perspectives are laid down to be the lighthouse for further research.

## 1. Introduction

Excessive emissions of greenhouse gases, such as carbon dioxide (CO_2_), have led to serious global environmental problems in recent years. Among them, the construction industry makes a huge carbon footprint, for which the CO_2_ emissions from cement production alone account for about 8% of total anthropogenic emissions [1]. As reported, the total global cement production in 2018 was 4.1 billion tons with an average global warming potential (GWP) and CO_2_ emissions of 1 kg cement production of 0.00123 and 981 kg CO_2 eq_, respectively [2]. Although lots of carbon-reducing and emission-reducing technologies are currently adopted by the building materials industry, carbon emissions remain at a high level mainly attributed to the growing demand for cement [3]. Against this background, the alkali-activated materials (AAM) prepared by aluminosilicate materials and alkaline activators have drawn much attention due to their low energy consumption and eco-friendly nature.

In the 1940s, Purdon prepared a new type of cement with high strength and rapid setting by using an alkali activator and slag [4], and he believed that sodium hydroxide (NaOH) was the key factor in the activation process. Subsequently, Glukhovsky [5] and Krivenko et al. [6] used various cementitious and activators, then found that aluminosilicate hydration products have extremely high stability, which laid the foundation for the development and application of the AAM. With the further development of research, the AAM can be roughly divided into two categories according to the reaction mechanism [7,8,9]: the high-calcium system with C-A-S-H gel-like tobermorite as the main polymerization product [10] and the low-calcium system with highly cross-linked and disordered N-A-S-H gel as the main polymerization product [11], which was later named geopolymers by Davidovits [12]. Compared with OPC, the AAM strength is mainly derived from polymerization rather than hydration, so it exhibits fast hardening and great early strength [13,14,15], acid and alkali resistance [16,17,18,19], and green environmental protection [20,21,22,23,24], and the potential commercial prospect of the AAM is also valuable. However, the brittleness [25,26], drying shrinkage [27,28,29], and other issues of the AAM also limit its promotion and development. On the other hand, the above the AAM is prepared by solutions of alkali metal hydroxides, silicates, and so on, and solid aluminosilicates in two parts involved in the reaction, and the preparation process requires the use of a large amount of highly corrosive and viscous chemical reagents, which poses potential difficulties for the transportation, storage, and construction of the AAM.

To address the issues of the AAM, a novel technology of one-part AAM (OP-AAM) obtain by “just add water” has been developed recently by mixing the precursor with the solid powder and then adding water only to produce [30]. The whole preparation process is similar to that of OPC, thus making it extremely convenient for construction users to produce on-site, which greatly increases the commercial value and feasibility of the AAM. However, there are still many technical problems in this preparation technology since its emergence, such as improving the precursor reaction activity, reasonably controlling the polymerization reaction rate to obtain the best setting time, adopting a suitable curing method for field mixing, and optimizing the ratio to enhance the mechanical properties and durability. In recent years, the OP-AAM has become the focus of attention. A large amount of literature has reported the use of new materials and their good performance. However, a systematic comparative analysis and performance evaluation are still lacking, studies of durability are also insufficient, and inconsistent conclusions between different scholars have also undoubtedly hindered its further promotion and application.

Based on the above discussions, this paper reports the latest research progress on the OP-AAM, summarizes, and collates the effects of precursors, solid alkali sources, admixtures, pretreatment processes, coordination ratios, fiber reinforcement, and the curing method. This work aims to provide a more comprehensive reference for the research and optimization of the OP-AAM and to promote the application of the AAM in production more effectively.

## 2. Raw Materials and Preparation

With the continuous development of the AAM, this alternative material of concrete with low-carbon, green, and high strength has gradually attracted the attention of scholars from various countries. Since the 1970s, alkali-activated materials have gone through several stages of development, and the preparation process and reaction mechanism have been progressively understood [31]. Figure 1 depicts the publication of the AAM in the past 20 years, in which the rapid development is obvious. It is in 2016 that a new preparation method of the AAM by “just add water” was first developed. Different from the conventional preparation process, this new production method overcomes the potential threats posed by the configuration and use of alkaline excitants and no transport and storage of liquid raw materials. As a result, the feasibility of production applications and safety is significantly enhanced. The comparison in Figure 2 shows that the AAM generates higher costs than that of OPC using conventional excitants. It should be noted that AAM may have low carbon emissions, but there are still some studies that show that a part of geopolymer concrete has similar or even higher environmental impacts than OPC due to the negative environmental impacts brought by sodium silicate production [32,33]. In addition, lower manufacturing costs may be achieved, if an alternative high-performance activator can be found. On the other hand, the quality control and production process of the precursor and activator used in the AAM have not yet formed a unified standard for evaluation, the development of related technologies and standard specifications is lacking, and a lot of research work is still needed for verification.

### 2.1. Aluminosilicate Precursors

At a high alkaline environment (pH > 11.5) [36], the Si and Al in the precursor are dissolved, and the 3D net structure has almost three forms generated by geopolymer reaction: polysialate (-Si-O-Al-O-), polysialate-siloxo (-Si-O-Al-O-Si-O-) and polysialate-disiloxo (-Si-O-Al-O-Si-O-Si-O-) [37]. Positive charge (such as Na^+^, K^+^, Ca^2+^, Fe^3+^) fills in the network to adjust the excess negative charge [38]. Depending on the Ca^2+^ concentration and pH value in the polymerization system, N-A-S-H gel and C-A-S-H gel can coexist or transform into each other [39,40], so the type of polymerization product depends on the different precursor systems.

The precursors used in the OP-AAM are the same as the conventional AAM; the difference only lies in the presence or absence of liquid components before preparation (Figure 3). Most studies have reported fly ash (FA) and ground granulated blast-furnace slag (GGBS) as precursors alone or in a mixture. Among them, the F-class FA is more widely used [41,42,43], because the C-class FA is not easy to meet the workability due to its high calcium content and rapid setting under the action of the activator [44,45,46]. The slag-based AAM, on the other hand, has higher activity and calcium content, so it tends to exhibit faster setting and higher early strength [47,48]. The Ca/Si ratio of C-A-S-H gel in a high-calcium system is lower than that of OPC, but due to the strong shrinkage of the slag-based AAM, it often needs to combine with other precursors to reduce the calcium content of the mixture [49]. Overall, the calcium content in the precursor fraction is beneficial for strength development, regardless of whether the material is an added calcium hydroxide or a high-calcium aluminosilicate. Since the particle shape of FA tends to be spherical, the paste often obtains great workability. The dissolution and reaction rate of FA is low at room temperature, and high-temperature curing is more useful to improve the polymerization efficiency [50]. Due to the platelike particle shape and high specific surface area of MK, it is easy to lead to poor slump [51], but MK obtains a lower shrinkage rate and longer setting time compared to the GGBS-based OP-AAM [52]. In addition, the fineness of the precursor particles likewise affects the mechanical properties of the sample. Other available precursor materials, such as nickel slag [53], lithium slag [54], metallurgical residues [55], silica fume [56], and rice husk ash [56], will also affect the mechanical properties of the AAM, as listed in Figure 4. These precursors can be roughly divided into two types of reactions. The part with higher calcium content forms C-A-S-H gel due to the combination of Ca^2+^, while the pozzolana materials are more inclined to form N-A-S-H gel. The two reaction modes have different effects on the performance of AAM. The hydraulic reaction is more likely to cure at ambient temperature, and it might cause high shrinkage and potential carbonization risk [57]. On the other hand, the pozzolan reaction type is less likely to decalcify to produce expansive products when resisting sulfate corrosion and is therefore more suitable for environments that contain erosive salts [58].

Clay minerals (hydrous aluminosilicates) typically require calcination in order to be reactive in alkali activation processes, and these materials have increased water demand due to their plate-like particle shapes, and the mixes are vicious. Kaolin (Al_2_Si_2_O_5_(OH)_4_) in its dehydroxylated form, metakaolin (Al_2_Si_2_O_7_), is one of the most studied aluminosilicate sources for geopolymer preparation [59]. Bentonite is usually composed of the clay mineral montmorillonite (Na,Ca)_0.33_(Al,Mg)_2_(Si_4_O_10_)(OH)_2_·nH_2_O. However, the thermal treatment at 550–850 °C is requested for alkali activations of bentonite to improve the reactivity [60]. Similar methods are also applied to dolomite [61], phyllite dust [62], red mud [63], and other mineral materials. Most importantly, the most suitable clay mineral material depends on the availability of local materials, when the feasibility and economy of material acquirement are accounted for.

Composite cement is generally the combination of the OPC and precursor materials. The most important feature of the composite cement is its hardening characteristics when a certain percentage (about 20%) of the OPC is blended with the precursor [61,64] and reduces the cost [65], which prepares low-carbon concrete materials by mixing AAM with cement a new application prospect.

### 2.2. Solid Alkali Sources

The distinction of the OP-AAM from the conventional AAM is discussed in this section. The critical components are from the solid alkali sources rather than the aqueous solutions of alkali metal hydroxides or the salt of strong alkali weak acid. That is, any substances that provide alkaline cations, raise the pH of the reaction mixture, and promote dissolution can be involved in the polymerization reaction as a solid alkali source [66]. The base source materials for integrated preparation include solid NaOH, Na_2_SiO_3_, Na_2_SiO_3_·5H_2_O, Na_2_CO_3_, NaAlO_2_, Na_2_SO_4_, KOH, red mud, and alkaline lime glass powder (see Table 1). It has been well noted that anhydrous sodium metasilicate (Na_2_SiO_3_, modulus 0.93) has higher compressive strength and better workability when the FA and GGBS are used [44,67,68]. In addition, quicklime (CaO), dolomite (CaMg(CO_3_)_2_), and slaked lime (Ca(OH)_2_) provide alkaline earth metal cations rather than alkali metal cations, which contribute to the formation of different binding phases in low-calcium systems [69]. The introduction of calcium sources in alkali-derived materials can increase the calcium content of the system and obtain higher mechanical properties [47] and may have an inhibitory effect on shrinkage due to the formation of the hydrotalcite phase [70]. Thus, for solid alkali sources, sodium silicate usually produces better properties, while the introduction of carbonate reduces the phenomenon of over-rapid setting and inhibits shrinkage due to solid sodium silicate [70], and sodium aluminate can slightly reduce the rheology of the slurry while keeping the slump steady [71].

The main difficulties limiting the widespread applications of the AAM are the preparation process safety and energy consumption. For example, the solid sodium hydroxide is corrosive, and exposure to carbon dioxide during transportation or storage can form sodium carbonate, leading to material deterioration. According to relevant statistics, NaOH is produced at a rate of approximately 60 Mt per year via the chlor-alkali process, and upscaling this production is not straightforward because chlorine (Cl_2_), which has a limited world market, is produced as a side product [99]. The production of alkali silicates also raises carbon emissions due to the high-temperature melting (850–1088 °C). It is thus important to seek a suitable material to replace alkali silicates. Sodium carbonate has similar effects to sodium hydroxide in the hydration reactions of metakaolin and bentonite [73,100] and can therefore be used as a potential alkali source material with carbonate exciters that have 80% lower carbon emissions than silicate cement [34]. A sodium aluminate produced by dissolving aluminum hydroxide in sodium hydroxide solution and treating bauxite at high temperature and pressure can provide active aluminum elements in the polymerization reaction, constituting a more dense matrix structure with enhanced compressive strength and drying shrinkage [101]. The use of precursors and activators in the existing literature is statistically presented in Figure 5. In addition, other activators, such as paper sludge [95], oyster shell powder [83], lime kiln dust [78], waste glass [74], coffee husk ash [102], fluorescent lamp waste glass powder [103], and biomass ash [84], are used for the preparation of solid alkali sources, but whether these can have effective scale production requires more experimental analysis to be tested.

### 2.3. Admixtures

Concrete admixtures are widely adopted to improve the workability, rheology, and mechanical properties of OPC. These admixtures include superplasticizers, retarders, antifreeze, and air-entraining agents. When the concrete workability is evaluated, the effects of the superplasticizer (such as lignosulphonate, naphthalene, melamine, and polycarboxylates) and retarders (like sugars, citric acid) are much more obvious. Because the extreme alkaline conditions of the AAM are different from those of OPC [68], these superplasticizers suitable for OPC may not suitable for the AAM [104]. As reported by Luukkonen et al. [105], the slag-based AAM with a sodium hydroxide activator, used lignosulphonate, melamine, and naphthalene superplasticizer are more effective than their counterparts with a polyacrylate and polycarboxylic superplasticizer. Oderji et al. [93], however, concluded that lignosulfonate admixtures are largely ineffective in terms of workability associated with a significant decrease in mechanical properties. It is suggested that 6 wt% borax can effectively improve workability. That is, systematic research should be carried out to find out the proper superplasticizers suitable for AAM [98,106]. Although the polycarboxylate superplasticizer may be one of the more effective admixtures reported in the studies, these performance differences may be affected by the precursor and the type of activator, so the use of a superplasticizer should be based on the actual situation.

In addition, due to the rapid polymerization reaction of high-calcium aluminosilicate materials, the short setting time leads to potential difficulties in casting in respect of the practical applications [107]. Limited research in terms of the retarders was carried out in integrated preparation techniques. Differently, most studies tried out to reduce the amount of high-calcium precursors. Moreover, fly ash [108], metakaolin [92], bentonite [60], natural pozzolan [88], red mud [63], tailings [67,77], and mineral materials [34] have been adopted to eliminate the calcium content of the mixed precursor as a way to inhibit the excessively short setting time. Further research was also conducted on the retarders. Wang et al. [53] used anhydrous citric acid as a retarder, which effectively prolonged the setting time of slag-based AAM. Oderji et al. [93] added 4–6 wt% borax to FA/GGBS-based AAM, and the workability of the paste was improved. Zheng et al. [72] used a partially carbonized CaO alkali activator, and the surface of the activator was modified by the thickened CaCO_3_ product layer, which increases the diffusion resistance to CO_2_ and can reduce the polymerization rate and prolong the setting time to some extent. Samarakoon et al. [35] mixed solid sodium hydroxide with lime glass powder, which effectively prolonged the setting time but reduced the fluidity significantly. Adesanya et al. [20] extended the setting time by adding micro silica powder; at the same time, the properties and efflorescence resistance were improved as well. Wang et al. [109] used calcium hydrogen phosphate as a retarder with a 7% replacement rate, providing better performance for the FA/GGBS-based AAM.

Overall, different admixtures used in the preparation process are to improve their fresh performance associated with better slump and setting time without affecting their strength. However, some concrete admixtures fail to obtain the desired performance, because of the highly alkaline conditions of the AAM. These mentioned admixtures are used to control the setting time and obtain better workability by preventing the aluminum and silicon from participating too quickly in the polymerization reaction or by reducing the dissolution rate of alkali components. Therefore, finding out the more effective admixtures either for a particular component or for the specific conditions is still the focus of the work in the coming period.

### 2.4. Preprocessing

At this stage, most studies favor preprocessing of precursors or activators to make them sufficiently reactive and improve polymer reaction efficiency. Table 2 counts several common preprocessing methods, in which the best activation is generally achieved by ball milling and calcination. For the OP-AAM, the alkali source (e.g., NaOH) is usually calcined to 600–1000 °C, and the solid crystalline phase is changed. This process is generally called the alkali melting. The optimal calcination temperature depends on the material used, such as bentonite calcines better at 1100 °C than 1200 °C. When dolomite or Na_2_CO_3_ is used as an activator, quartz, usually as an impurity component, becomes a reactive glassy phase after calcination with NaOH at 1000 °C [60]. This calcination treatment is also applicable to kaolinite, which carries out dehydroxylation and loss of its long-range ordered structure, the conversion of the powder to an amorphous form as metakaolin [88], whose calcination with Na_2_CO_3_ or NaOH has a positive effect on mechanical properties when the temperature is increased to 950 °C. Figure 6 shows a special chemical treatment method to solidify alkali solution effectively, which is a new attempt to apply solid alkali source materials.

However, the calcination of raw materials increases carbon emissions and energy consumption. As a result, the low-carbon advantage of the AAM was correspondingly affected. In some cases, carbon-containing substances, such as dolomite or Na_2_CO_3_ added during the calcination process, will also lead to CO_2_ emissions. Therefore, it is possible to reasonably save energy and improve the efficiency of solid waste utilization by properly selecting aluminosilicate materials (such as FA and GGBS).

### 2.5. Mix Proportion

Different from these complex parameters considered in the design of the traditional AAM, the water–binder ratio (or liquid–solid ratio), alkali–binder ratio, precursor composition (such as GGBS/FA), molar ratio of major elements (such as SiO_2_/Al_2_O_3_, Na_2_O/SiO_2_), and admixtures are critical parameters to be considered for the OP-AAM. The effects of various parameters on workability and mechanical properties will be introduced in Section 3.

When the elemental content is evaluated, the molar ratio of SiO_2_/Al_2_O_3_ for the traditional AAM achieves better results in the range of 3.3 to 4.5. Meanwhile, the molar ratio ranging from 0.75 to 6.02 is much more suitable for the OP-AAM. Hajimohammadi et al. concluded that the SiO_2_/Al_2_O_3_ ratio of metakaolin-based AAM showed a decreasing trend in compressive strength when it was increased from 3.5 to 4.5 [111]. However, Sturm et al. suggested that increasing the SiO_2_/Al_2_O_3_ molar ratio increases the compressive strength and acid resistance [94]. A possible explanation for this different observation is the breakage of the silica–oxygen bond during dissolution in an acidic environment. When the reduction of aluminum elements in the polymer backbone stays at the higher Si content, the partially released SiO_2_ into the solution protects by the formation of silica precipitates on the surface, as confirmed in Luukkonen’s study [56]. This suggests that the tendency of the compressive strength to vary with the SiO_2_/Al_2_O_3_ ratio is not constant in the material and depends on the different precursors and activator types. Although the alkaline condition with a sufficiently high concentration to dissolve the aluminosilicate is requested, the threshold range must be carefully controlled. It is mainly because the excessive alkali concentration increases the risk of efflorescence (expressed as M_2_O/Al_2_O_3_ molar ratio, where M stands for Na or K, whose optimum value should be close to 1) [99]. A research conducted by Peng et al. [60] verified that the mechanical properties of the resulting samples were significantly lower if no alkali component was added during the calcination of bentonite, and at this limit, Coppola et al. confirmed that the AAM in an aggressive environment durability studies likewise proved this pattern [90].

The water requirement (i.e., the amount of water required to form a standard consistency paste) of a different precursor is generally influenced by variable factors (e.g., the particle size, shape distribution, and specific surface area). Moreover, the heat generated by the dissolution of the solid activator may result in significant water loss [112]. As previously investigated, the proper water–binder ratio is roughly within the range of 0.16 to 0.6, which can be adjusted to obtain acceptable workability. In particular, the water consumption is roughly inversely proportional to the mechanical properties under the premise of ensuring workability. With the shortage of fresh water, seawater and treated wastewater have also been selected as potential alternatives. It was found that the AAM with seawater exhibited an accelerated activation process and featured a short setting time and good mechanical properties [85]. As reported by Luukkonen et al., the application of reverse osmosis rejects water, reducing the porosity and improving the strength at the long age of the AAM. Even though these researchers do provide a tentative idea for the water treatment [86], whether the high chloride ion content of seawater and other corrosive media will affect the durability performance of the AAM or corrode the steel under the reinforcement concrete is still unknown.

### 2.6. Fiber Reinforcement

Table 3 shows the differences of performance with various fiber types. Abdollahnejad et al. [43,108,113] studied the various properties of steel fibers, polypropylene fibers (PP), and polyvinyl alcohol fibers (PVA) and the durability performance of the fiber reinforcement AAM under freeze–thaw cycles and corrosive conditions. In the latest studies [62,114], basalt and glass fibers were also added, and it was found that the reinforcing effect of different fibers is not consistent with different precursors. The best overall performance was at 1 vol% replacement of steel fiber, and the fiber length has a positive benefit to the improvement of load-carrying capacity. At a suitable volume content, the fiber can show a positive behavior of compressive strength, while exceeding a certain threshold may lead to the weakening of mechanical properties due to the generation of a weak interfacial transition zone (ITZ) [114]. On the other hand, mineral fibers have unstable performance in a strong alkali environment and easily lead to strength loss. Under the premise of ensuring that the fibers are available, the addition of fibers improves the mechanical properties of paste, especially that the flexural strength and elastic modulus improvement is significant [115], but the case of corrosion and destruction of mineral fiber materials with poor alkali resistance should be considered when they are used, or coating materials can be developed to protect the fibers from reactions.

### 2.7. Curing Method

Being cured at either the ambient temperature (about 20 °C) or the high temperature (40–80 °C) mainly depends on the precursors, mix design, and so on of the AAM. Table 4 lists several feasible curing conditions, and each method also has a slightly different applicable condition. Note that the heat generated by the solid activator during the dissolution process may be a benefit to the curing process [112]. If there is no water evaporation or alkaline leaching during the curing process, the temperature will only affect the reaction rate rather than the final products. In addition, the early strength increases with temperature, and the long-term strength will gradually converge [41]. Although the low temperatures may not be conducive to the hydration reaction, the strength of the slag-based AAM at 28 d is 68 MPa when it was cured at −5 °C. Note that the OPC under the same conditions is only 9.7 MPa [81]. Moreover, the slag-based AAM can continue its hydration process when it was transferred from the subzero temperature to the normal temperature [55], being an important finding for the preparation of the AAM in a cold region.

In addition to the temperature, relative humidity is another critical factor to consider when the curing method is evaluated. Under constant humidity curing conditions or sealing curing, the density of the AAM paste can be improved and the pore space will be reduced. Dehydration during the curing process can lead to incomplete hydration reactions, accompanied by efflorescence and microcracking, making the strength decrease [116]. Although it is important to maintain the ambient humidity, it is a controversial issue whether to adopt water curing for the AAM. Wei et al. found that paste would be affected by ion leaching in water curing, which may damage the microstructure and reduce hydration products [73]. In contrast, Haruna et al. verified that the water curing did not result in strength loss and that its strength development was essentially the same as that of the standard curing method, and they also found that solar curing was even stronger than standard curing, which may be a promising curing method in the future [89].

Based on the above discussion, the curing method is an important factor for the OP-AAM. After the mixture of the solid activator and the fresh water, the heat released will accelerate the evaporation of water. However, using water curing (immediate immersion in water) can dilute the activator and reduce the efficiency of polymer generation [117]. Sealing with the plastic cover is believed to be an effective method to prevent water loss during the polymerization reaction. The fly-ash-based AAM may be more suitable for high-temperature curing if the curing condition is eligible.

## 3. Properties of One-Part AAM

### 3.1. Workability

The fresh performance of the OP-AAM is mainly evaluated by the slump and setting time. As depicted in Figure 7, the alkali–binder ratio (a/b ratio), water–binder ratio (w/b ratio), and GGBS content do affect the workability of the AAM. The most sensitive parameter is the w/b ratio. With the increase in water consumption, both the slump and setting time are enhanced with the reduced mechanical properties. It can also be found from Figure 7 that the hardening properties of the paste are mainly affected by the value of the a/b ratio. Although the setting time of the AAM is further reduced as the alkali content increases, the increased a/b ratio will affect the flowability. The gels of polymeric products depend on the dissolution of precursors. In a highly alkaline environment, Si and Al are dissolved and polymerized to form hydraulic binders [119]. Increasing the alkali content will accelerate this process [120]. Different from the positive results reported by lots of scholars, Teo et al. [44] and Li et al. [64] found a rapid decrease in slump when the alkali equivalent exceeded a certain limit.

As mentioned before, many materials can be incorporated into the mix design, and their performance varies. In this section, we take the slag, for example, in comparing the workability of the OP-AAM with different slag replacements. It can be seen that the properties of the AAM are different when the slag is added to different precursors. Most research showed that the slag-based AAM obtains fast hardening properties, resulting in a decrease in slump and setting time at a higher slag replacement [121]. As listed in Figure 7, the slump tends to increase with higher slag content, while the setting time is consistent with the conventional pattern. The possible reason for this finding may be attributed to the higher reactivity of the slag when light-burnt dolomite (LBD) is used as a precursor. The active Ca and Mg components in LBD generate calcium hydroxide, C-A-S-H gels, and magnesium hydroxide, which will affect the reaction heat of the slag-based system [122]. In contrast, Chen et al. [91] using lead–zinc mine tailings and Shah et al. [54] using lithium slag blended into a slag-based AAM found that higher fineness needs higher water requirement; thus, the slump loss is due to the fineness, not the reactivity [123].

In addition, the evaluation of workability has different parameters specified in different standards, such as the use of European standards EN 1015-3 and EN196-3 [80] and Indian standard IS 1199–1959 [124]. These standards use a minislump test to characterize workability, which cannot be evaluated uniformly with the above index and is not listed in this paper to facilitate the comparison of similar parameters.

### 3.2. Rheology

The material resistance to flow after the material begins to flow and the shear stress required to initiate flow, namely, plastic viscosity and yield stress, respectively, are important parameters in the rheological properties of the AAM [125]. For the one-part AAM, when the water content is determined, increasing the dosage of the activator may reduce the yield stress [126]. Owing to the influence of ionic equilibrium in the dissolution process of the solid activator, the dissolution rate becomes slow when the solution is close to saturation, and the equilibrium shear stress is reduced. Generally speaking, the rheological law of the OP-AAM is similar to that of the traditional AAM. The basicity of the solid activator after dissolution affects the dissolution efficiency of the precursor, and the alkali cation mainly plays the role of charge balance [120]. Among the rheology of the traditional AAM, the sodium silicate activator may have a lower yield stress, which is due to silicate oligomers adsorbing on the surfaces of slag particles, resulting in insufficient dissolved Ca^2+^ in the solution and greater repulsive double-layer forces [126]. However, when sodium silicate is added to a solid phase, the yield stress and plastic viscosity could be increased [127]

In addition to the activator, aluminosilicate also has a great influence on rheological behavior. Different from GGBS, it dissolves faster and has higher reactivity in alkaline environments [128,129]. The “ball-bearings” and slow dissolution rate of FA make gel precipitate slower at the initial stage of the reaction [129], reducing the yield stress of pastes and affecting its rheological [130,131]. On the contrary, Alrefaei et al. concluded that the addition of FA can increase the yield stress of the OP-AAM, which may be related to the high porosity and surface area of carbon-containing impurities [126]. MK generally exhibits high viscosity and yield stress, probably due to the platelike particle shape and high surface area [119,132]. It means that a suitable activator and enough water are needed to ensure acceptable performance [132,133].

### 3.3. Compressive Strength

According to the statistics in Table 4, being the main precursor used for the AAM, the application of the slag can provide the highest compressive strength of all mix designs [48,71]. This is mainly due to its calcium content and the C-S-H gel generated by the hydration reaction, the latter of which greatly improves the mechanical properties [81]. The incorporation of other precursors in slag-based AAM can retard the setting time [41,64,108] and reduce the drying shrinkage [61,101,134], which can meet the workability requirement. Under specific conditions (e.g., high curing temperature, chemical modification), the fly ash or metakaolin can also achieve satisfactory mechanical properties [71,91,102], providing a variable precursor to be selected. In terms of alkali sources, sodium silicate is most effective, followed by sodium hydroxide [67,73]. The performance of other alkali sources, such as carbonates [47], aluminates [71,94], calcium oxides [21,77], and potassium oxides [90,134], has been successfully used for preparation. When these materials are blended with water, the alkali environment as well as aluminum and silicon elements for the hydration reaction may be generated.

It can be seen from Figure 8 that the compressive strength gradually increases with the values of the a/b ratio and GGBS content. The possible reason for this observation is that the increased a/b ratio within a certain range (generally in the range of 5–12%) can promote the hydration reaction. Meanwhile, there is a risk of efflorescence on the surface [79], forming pores and reducing the compactness [135] when it is at high concentrations. The effect of the w/b ratio on the compressive strength of the AAM is negative, which agrees well with previous research on other cementitious materials.

The effect of the Si/Al ratio can also be seen in Figure 8. Increasing the SiO_2_/Al_2_O_3_ molar ratio positively contributes to compressive strength and modulus of elasticity, together with a decreased porosity [136]. Meanwhile, the unreacted materials decrease with increased alkali and silicon content [59]. In contrast, Adesanya et al. [20] found that the use of silica fume to increase the soluble silica along with desulfurization dust (DeS) resulted in better mechanical properties than sodium hydroxide. However, this pattern was not monotonically increasing, reaching up to 33.6 MPa at Si/Al = 6.36. This induced change in mechanical properties could be attributed to the fact that Fe or Ca elements for additional dissolution and the beneficial effect of Fe in AAM have been demonstrated by Simon et al. and Adesanya et al. [137,138].

If only superior mechanical properties are requested, using highly reactive materials and improving the a/b ratio and Si/Al ratio are believed to be effective methods. However, the fact is that the fine-grained materials usually require much more water to meet the workability, and the higher w/b ratio brings a side-effect on the mechanical response. That is, developing an effective superplasticizer for the AAM becomes critical in balancing the mechanical and physical properties.

### 3.4. Durability

The durability of the AAM prepared by different precursors is different. Although the AAM exhibits superior resistance to carbonation and sulfate attack, the freezing resistance of which has not been well evaluated. Alzaza et al. [55] found that the incomplete hydration and internal deterioration led to significantly lower freezing resistance when it was curved at −20 °C. As a result, increasing the curing temperature is beneficial to form a dense microstructure and improve the freezing resistance. Coppola et al. [90] considered the effect of the durability of the slag-based AAM under the influence of an air-entraining agent (AEA), the addition of AEA improved the freezing resistance of all tests, the test groups with high alkalinity (AAS12, AAS16) could achieve freezing resistance similar to that of OPC, and slag-based AAM showed higher corrosion resistance to the CaCl_2_ condition than the MgSO_4_ condition, which can be attributed to the decalcification of the C-S-H gel and the formation of the expansion product gypsum. Ahmad et al. [92] investigated the performance of a GGBS/FA-based AAM against sulfate and acid corrosion under different mineral admixtures. All their samples were subjected to up to 270 d of age immersion in sodium sulfate solution with only up to 17% loss of strength. It has been found that there is better resistance to sulfate attack (1% loss of strength) at 4.5% metakaolin replacement, while samples in a sulfuric acid environment showed varying degrees of cracking and spalling, which was also observed in acid resistance experiments by Sturm et al. [94].

Existing studies on the durability of the OP-AAM are extremely limited, and it appears that slag-based materials may deteriorate in performance under sulfate attacking due to expansive products, which is an obstacle to be overcome. Excellent resistance to carbonation and high temperatures has been demonstrated by several studies. However, the deficiency of freezing resistance is not good enough, which is similar to the conventional AAM [17,139]. This does not prove whether there is a performance difference in the AAM prepared by traditional alkali solutions. As a result, more studies are requested to clarify both the similarities and differences of the AAM with two typical preparation techniques. The investigation results will be extremely important to better understand the service performance of the AAM in harsh environments.

## 4. Conclusions and Future Research Need

In this study, a comprehensive literature review of the mechanical properties, chemistry, raw materials, and reaction mechanisms of one-part alkali-activated materials (OP-AAM) was conducted. It provides a clear understanding and guidance for developing future product developments in field applications of the OP-AAM for civil engineers and the industrial community. Here is a concise overview:Ground granulated blast-furnace slag (GGBS) and fly ash (FA) are the two most popular raw materials in the precursors. GGBS obtains the highest mechanical strength attributed to its excellent reactivity. However, the high shrinkage and low flowability should be well considered. The excellent mechanical properties of either the GGBS or the FA are still the best choice at present.Sodium silicate (SS) and sodium hydroxide (SH) alone or in combination can form a good performance, but SH poses some safety risks. The alkali content of the mixture can be increased by other alkaline earth metal hydroxides (e.g., CaO, Ca(OH)_2_). Moreover, some treated solid wastes (e.g., paper sludge, oyster shell powder, biomass ash) can be used as the potential alkali source materials.Polycarboxylic is an efficient superplasticizer for the AAM compared with its counterparts. Other precursor components, such as lignosulphonate, naphthalene, and borax, can also be beneficial to certain materials.Different from the pretreatment of raw materials by calcination and ball milling to improve their reactivity, the effects of the chemical activation or changing water temperature are not always obvious. A new treatment method with the additional delayed components can artificially regulate the reaction process and minimize the competitive adsorption phenomenon.Considering the high alkali equivalent of the AAM, these mineral fibers (basalt fibers, glass fibers, etc.) are not highly recommended. However, the stability of the steel fibers in acid and alkaline conditions is relatively constant. Compared with a single fiber, the mixed fibers may overcome some defects of the AAM. Regardless of the fiber type, fiber reinforcing has a better contribution to the tensile and flexural properties of the AAM.Normal temperature sealing is an effective curing method for slag-based AAM. For the fly ash-based AAM, a high temperature is recommended if the high performance of the AAM is requested. The other effective method is to add different materials into the binary or multiple precursors. Note that the water bath curing may lead to a significant strength reduction attributed to the alkali leaching.For the workability and rheological behavior of the OP-AAM, the solid activator may produce a different behavior from the liquid activator, and the dissolution heat may accelerate the hydration reaction and increase the yield stress. However, it is also noted that the evaporation of water makes the hydration process inadequate.The AAM has a high performance in carbonation resistance and high-temperature resistance. For acid resistance or sulfate resistance, slag-based AAM is more affected by expansion products, such as gypsum, due to its higher calcium content, and proper blending of fly ash and metakaolin can improve sulfate erosion resistance. However, the freezing resistance of the AAM should be well investigated.The main features of the OP-AAM are its cost-effectiveness and safe working compared with the traditional AAM. It is thus feasible to cast the AAM on-site with the one-step preparation process. Nonetheless, the commercialization of this technology has not yet been fully developed. Hopefully, these alkali-activated or geopolymer materials may be more widely used in the construction industry in the future.

## Figures and Tables

**Figure 1 polymers-14-05046-f001:**
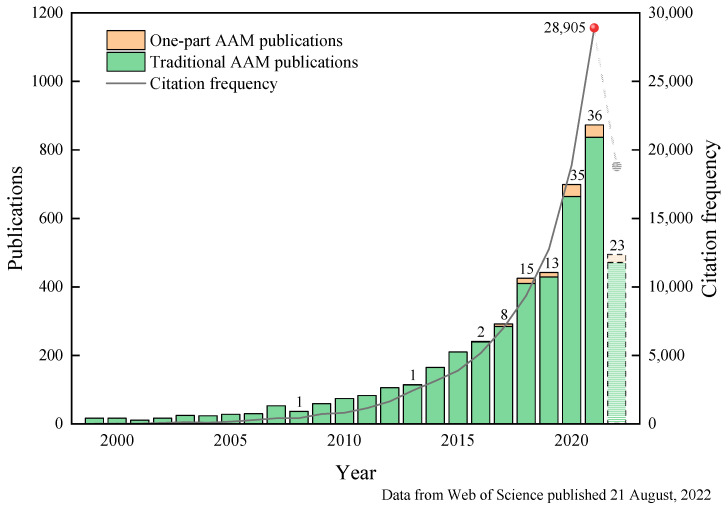
Publication and citation frequency statistics of traditional AAM and one-part AAM during 1999–2022.

**Figure 2 polymers-14-05046-f002:**
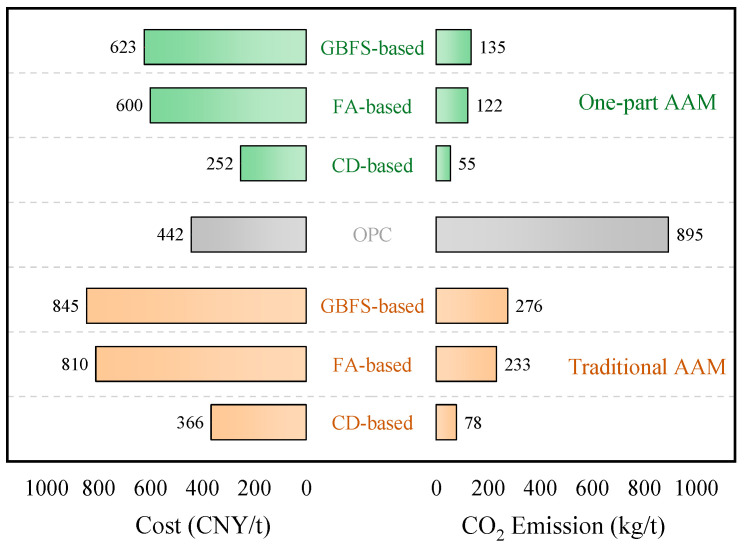
Cost and carbon footprint of AAM and OPC [34,35].

**Figure 3 polymers-14-05046-f003:**
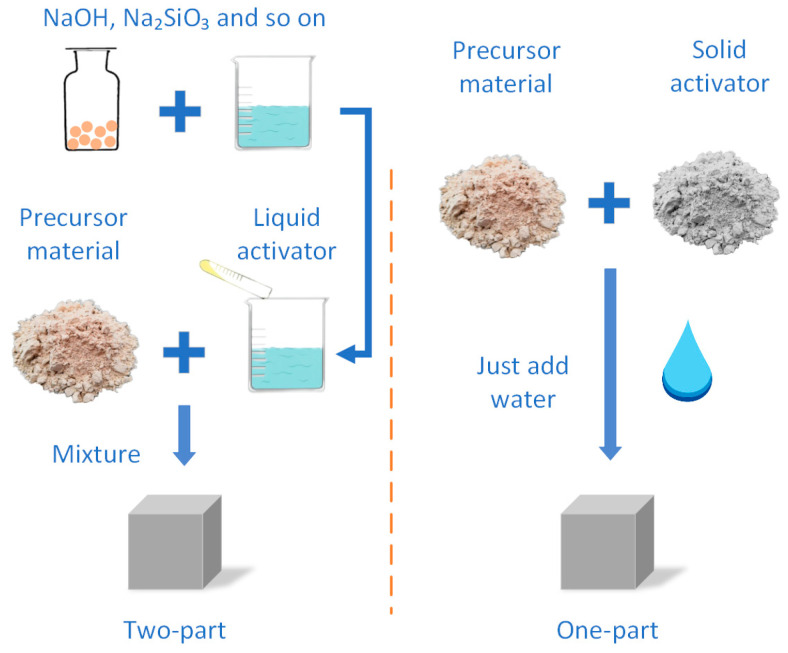
Schematic representation of different preparation processes.

**Figure 4 polymers-14-05046-f004:**
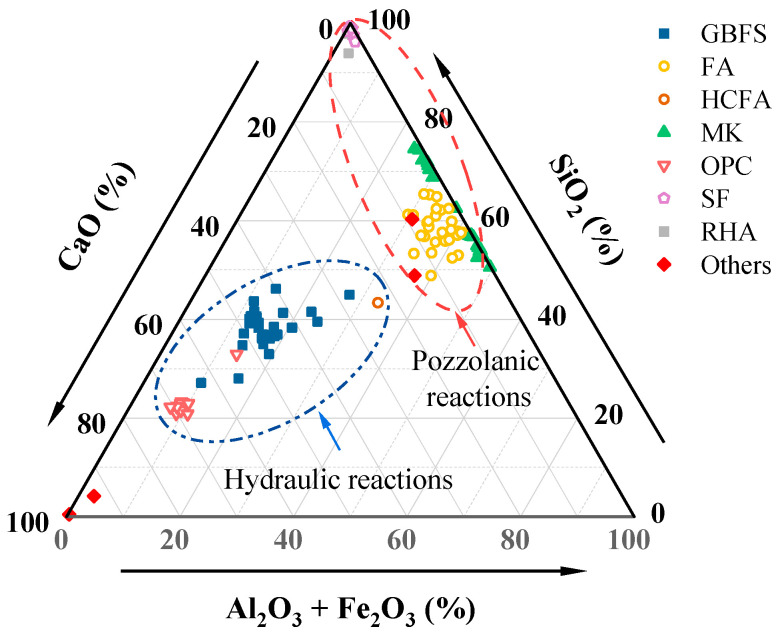
Ternary diagram of the main precursor.

**Figure 5 polymers-14-05046-f005:**
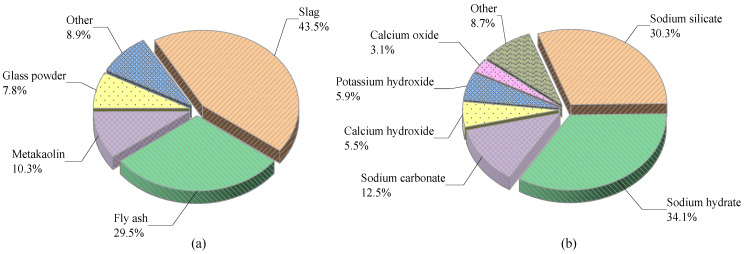
Statistics on the use of AAM (**a**) precursors and (**b**) activators.

**Figure 6 polymers-14-05046-f006:**
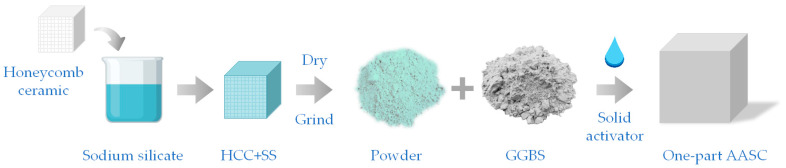
A novel technology for producing an alkaline activator using porous honeycomb ceramics (HCC) [87].

**Figure 7 polymers-14-05046-f007:**
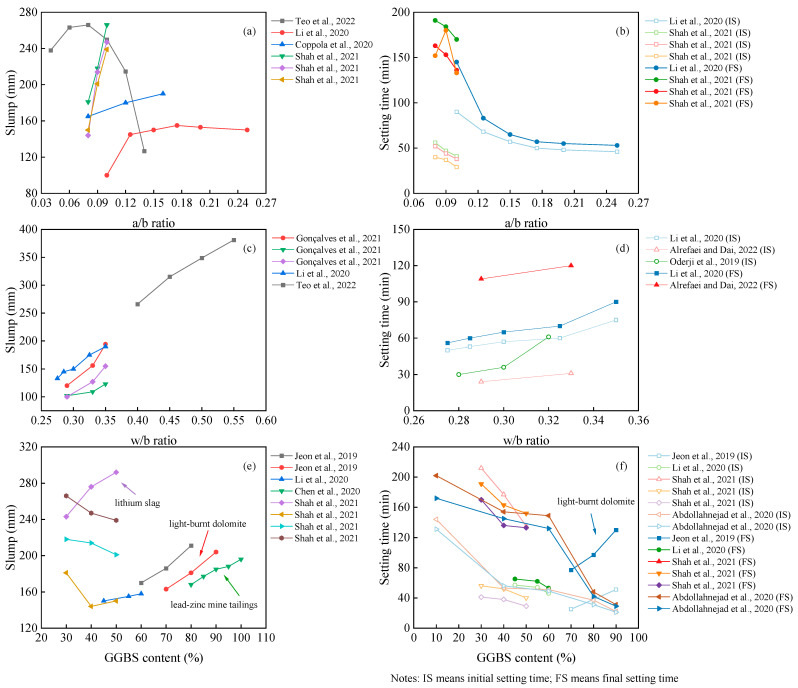
Influence of a/b ratio (**a**,**b**); w/b ratio (**c**,**d**); GGBS content (**e**,**f**) on OP-AAM performance [44,54,61,64,68,80,90,91,93,108].

**Figure 8 polymers-14-05046-f008:**
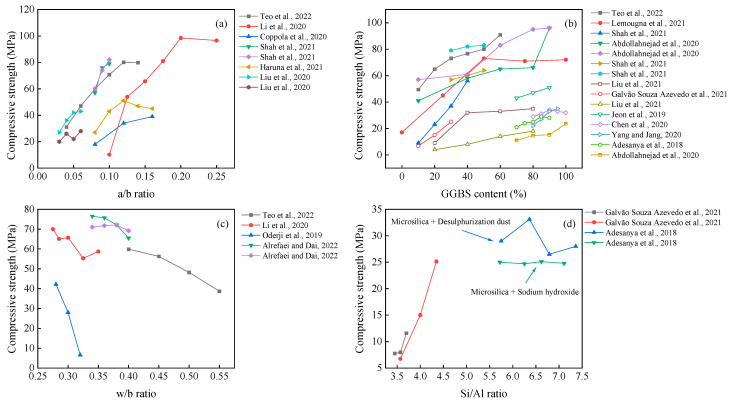
Effects of a/b ratio (**a**); GGBS content (**b**); w/b ratio (**c**); Si/Al ratio (**d**) on compressive strength of one-part AAM. [42,44,54,59,61,64,67,68,79,83,87,90,91,93,95,108]

**Table 1 polymers-14-05046-t001:** Performance of one-part AAM with different precursors and activators.

Author	Precursor	Activator	w/b Ratio	Slump (mm)	28 d UCS/MPa
Wang et al. [53]	NS	Na_2_SiO_3_	0.35	-	21–53
Teo et al. [44]	GGBS + FA(C)	Na_2_SiO_3_	0.4	196–214	49–91
Alrefaei and Dai [68]	GGBS + FA	Na_2_SiO_3_	0.42	-	49–63
Zhou et al. [47]	GGBS	Na_2_CO_3_ + Ca(OH)_2_	0.46	-	31–36
Zheng et al. [72]	GGBS	Na_2_CO_3_ + CaO	0.4	-	45–48
Wei et al. [73]	GGBS + FASB	Na_2_SiO_3_, Na_2_CO_3_	0.35	-	42–76
Wang et al. [71]	GGBS + FA	Na_2_SiO_3_, Na_2_CO_3_, NaAlO_2_	0.36	205–249	68–95
Samarakoon et al. [74]	GGBS + FA + WS	NaOH	0.5	-	21–28
Ren et al. [75]	GGBS	Na_2_SiO_3_	0.4	235–239	58–67
Refaat et al. [76]	GGBS	NaOH	0.27–0.31	60–150	9–53
Perumal et al. [48]	GGBS + SF + PD	Na_2_SiO_3_	0.25–0.35	-	46–145
Liu et al. [77]	GGBS + GMT	NaOH + CaO	0.5	-	14–45
Liu et al. [42]	GGBS + GMT	NaOH	0.4	-	4–36
Lemougna et al. [67]	GGBS + CS	Na_2_SiO_3_	0.32–0.4	-	28–76
Kadhim et al. [78]	MK + NP	LKD	0.45	-	6–20
Haruna et al. [79]	FA	Na_2_SiO_3_	0.25	-	46–68
Gonçalves et al. [80]	GGBS	Na_2_SiO_3_	0.29–0.35	102–196	41–56
Galvão Souza Azevedo et al. [59]	FA + WS/MK	NaOH + Na_2_SiO_3_	0.6, 0.9	-	5–12
Alzaza et al. [81]	GGBS	Na_2_SiO_3_	0.35	-	15–101
Alzaza et al. [55]	GGBS + SMP	Na_2_SiO_3_	0.35	-	3–43
Almakhadmeh and Soliman [82]	GGBS	Na_2_SiO_3_	0.4	190	77–89
Ali Shah et al. [54]	GGBS + LS	Na_2_SiO_3_	0.3	101–285	9–56
Yang et al. [34]	GGBS + CD	Na_2_CO_3_	0.6	-	27–45
Yang and Jang [83]	GGBS	COS	0.4	-	22–35
Samarakoon et al. [35]	GGBS + FA	SLGP + NaOH	0.4	-	28–45
Mobili et al. [84]	MK	BA	0.49–0.65	-	1.6–3.7
Lv et al. [85]	GGBS + FA	NaOH + Na_2_SiO_3_ + Na_2_CO_3_	0.45	-	41–44
Luukkonen et al. [86]	GGBS	Na_2_SiO_3_	0.35	-	70–90
Liu et al. [87]	GGBS	Na_2_SiO_3_ + HCC	0.5	-	21–44
Li et al. [64]	GGBS + WS + CAC	Na_2_SiO_3_	0.275–0.35	70–175	46–70
Kadhim et al. [88]	MK + NP	LKD	0.55	-	9–27
Haruna et al. [89]	FA(C)	Na_2_SiO_3_	0.25	34–165	41–70
Coppola et al. [90]	GGBS	Na_2_SiO_3_ + KOH + Na_2_CO_3_	0.55	150–220	14–48
Chen et al. [91]	GGBS + LZMT	Na_2_SiO_3_	0.45	168–191	29–34
Ahmad et al. [92]	GGBS + FA + MK/SF/MgO/OPC	Na_2_SiO_3_	0.16	223–305	28–41
Oderji et al. [93]	GGBS + FA	Na_2_SiO_3_	0.3	231–278	8–43
Ababneh et al. [21]	MK	Na_2_SiO_3_ + CaO + Na_2_CO_3_	0.4–0.53	-	7–20
Jeon et al. [61]	GGBS + CD	Na_2_SiO_3_	0.4	165–218	28–52
Sturm et al. [94]	GGBS + SF/SiO_2_/RHA	NaAlO_2_	0.38–0.5	-	30–58
Luukkonen et al. [56]	GGBS + SF/RHA	Na_2_SiO_3_	0.35	-	30–107
Adesanya et al. [95]	GGBS	NaOH + PS	0.31	-	33–42
Abdel Gawwad et al. [70]	GGBS	NaOH + MgCO_3_	0.3	-	56–83
Peng et al. [60]	CB + CD	Na_2_CO_3_	0.35	-	18–38
Almalkawi et al. [96]	VP	CaO + Na_2_SO_4_ + Na_2_CO_3_	0.5	-	5–22
Ye et al. [97]	RM + FA	NaOH	0.5	-	1.1–1.8
Alrefaei et al. [98]	GGBS + FA	Ca(OH)_2_ + Na_2_SO_4_	0.27–0.4	-	21–74

Notes: GGBS means ground granulated blast-furnace slag, FA means fly ash, MK means metakaolin, NS means nickel slag, FA(C) means C-class fly ash, FASB means fly ash sinking beads, WS means waste glass, SF means silica fume, PD means phyllite dust, GMT means gold mine tailing, CS means copper slag, NP means natural pozzolan, SMP means submicron particle, LS means lithium slag, CD means calcined dolomite, CAC means calcium aluminate cement, LZMT means lead–zinc mine tailing, RHA means rice husk ash, CB means calcined bentonite, VP means volcanic pumice, RM means red mud, LKD means lime kiln dust, COS means calcined oyster shell, SLGP means soda-lime glass powder, BA means biomass ash, HCC means honeycomb ceramic, PS means paper sludge.

**Table 2 polymers-14-05046-t002:** Common treatment processes and their evaluation.

Type	Treatment	Parameter	Major Process	Evaluation	References
Mechanical	Ball milling, crushing	Ball loading rate; ball material ratio	Increase the contact area and improve reactivity by mechanical treatment to accelerate the destabilization process of aluminosilicate structure	Highest reactivity and easy to handle	[53,101,110]
High temperature	Calcination	Temperature; duration	High temperature changes the mineralogical phase, loss of its long-range ordered structure and transformed into an amorphous form, improving the crystallinity and reactivity	High reactivity but increases carbon emissions	[59,83,88,96,101]
Chemical	Immersion, thermochemical, adsorption	-	The crystal original structure has been destroyed, which is conducive for the dissolution of silicon and aluminum	Medium reactivity, need to control chemical treatment level and not easy to use	[42,72,76,87]
Delayed addition	Delayed adding PCE	Delay time	Minimized the competitive adsorption phenomenon	No change in reactivity, slight increase in strength	[68]
Mixing process	Water temperature	Hot water, cold water	High temperature could speed up the initial reaction or reduce setting time; cold water can offset the early thermal shrinkage	Low reactivity, but provides controlled field mixing conditions	[82]

**Table 3 polymers-14-05046-t003:** Fiber type on the performance of one-part AAM.

References	Fiber Type	Dosage of Fiber (Vol.)	Compressive Strength	Flexural Strength	Effect Evaluation
Abdollahnejad et al. [113]	ST	0.5% and 1.0%	ST: 53–64 MPa	n.r.	ST can get the best mechanical properties and has the highest chemical stability, while PP/PVA may have chemical adhesion.
	PP		PP: 45–60 MPa		
	PVA		PVA: 47–61 MPa		
Abdollahnejad et al. [108]	PP	1.5%	PP: 50 MPa PVA: 56 MPa BA: 52 MPa Hybrid: 42–61 MPa	PP: 7.8 MPa PVA: 10.7 MPa BA: 11.3 MPa Hybrid: 7.8–10.9 MPa	All groups showed a decreased compressive strength from the original state; PVA has a better mechanical performance.
	PVA				
	BA				
Shah et al. [114]	ST	0.5%, 1.0%, 1.5% and 2.0%	ST: 28–33 MPa	ST: 2.8–3.1 MPa	The strength decreased when the dosage of the fiber was beyond 1.5%; BA has bad adhesion with the OP-AAM.
	PVA		PVA: 25–30 MPa	PVA: 2.8–3.2 MPa	
	BA		BA: 29–31 MPa	BA: 2.3–2.7 MPa	
Abdollahnejad et al. [43]	ST	1.0%	ST: 82 MPa PVA: 76 MPa BA: 33 MPa CEL: 53 MPa Hybrid: 54–72 MPa	ST: 10.7 MPa PVA: 9.6 MPa BA: 6.1 MPa CEL: 5.2 MPa Hybrid: 6–11 MPa	Except for ST, all single and hybrid fibers increased porosity and water absorption; ST can improve both fire resistance and freezing resistance.
	PVA				
	BA				
	CEL				
Perumal et al. [62]	ST	1.0%	ST: 150–233 MPa	n.r.	ST outperformed other fibers in the AAM, and longer the fiber, the better was the load-carrying capacity. Mineral fibers were unstable in high alkaline environment.
	GLA		GLA: 143–177 MPa		
	BA		BA: 135–168 MPa		
Alrefaei and Dai [115]	ST	0.5%, 1.0%, 1.5% and 2.0%	ST: 60–81 MPa	ST: 6.6–8.2 MPa	PE showed a comparative modulus of rupture relative to that of ST, and ST showed minor effects on flexural cracking and ultimate strengths.
	PE		PE: 45–63 MPa	PE: 7.3–8.0 MPa	

Notes: ST means steel fiber, PP means polypropylene fiber, PVA means polyvinyl alcohol fiber, BA means basalt fiber, CEL means cellulose, GLA means glass fiber, PE means polyethylene fiber, n.r. indicates data not reported.

**Table 4 polymers-14-05046-t004:** Comparison of different curing methods on one-part AAM.

References	Curing Method	Mix Parameter	Major Results	Evaluation
Abdollahnejad et al. [113]	Sealing/water curing	GGBS/PCW + Na_2_SiO_3_	Water curing strength +3%–+13%	Water curing has better properties for compressive strength and flexural strength
Abdollahnejad et al. [118]	Thermal curing	GGBS/PCW (porcelain and raw) + Na_2_SiO_3_	Compressive strength −20%–+25%	Thermal curing led to the maximum improvement of strength
Ahmad et al. [92]	Ambient/water/thermal curing	FA/GGBS + Na_2_SiO_3_	Water curing strength −92–−60%; thermal curing strength +5%–+135%	Water curing had an adverse effect
Haruna et al. [89]	Ambient/solar/water curing	FA(C) + Na_2_SiO_3_	Water curing strength –3%–−7%; Solar curing strength +6%–+23%	Solar curing had the best compressive strength; ambient curing had nice flexural strength
Alzaza et al. [81]	Subzero curing (−5/−10/−20 °C)	GGBS + Na_2_SiO_3_	Subzero curing strength –45%–−86%	The reactivity of AAS at low temperature is better than that of OPC; the strengths of AAS and OPC are 68 and 9.7 MPa, respectively
Alzaza et al. [55]	Subzero curing (−5/−10/−20 °C)	GGBS/SMP + Na_2_SiO_3_	Compressive strength under −5 °C is much higher then −10 and −20 °C	Low temperature hinders the reaction process, and adding SMP can obtain a better performance under subzero curing
Wei et al. [73]	Ambient/water curing	FASB + Na_2_SiO_3_/Na_2_CO_3_	Water curing strength –14%–−20%	Water curing decreases the quantity of hydration products and makes the microstructure much coarser

Notes: PCW means porcelain ceramic waste, GGBS means ground granulated blast-furnace slag, FA means fly ash, NS means nickel slag, FA(C) means C-class fly ash, SMP means submicron particle, FASB means fly ash sinking beads.

## Data Availability

All necessary data are provided in the article.

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
