# Peer review of "One-Part Alkali-Activated Materials: State of the Art and Perspectives"

_polymers, 2022, doi:10.3390/polym14225046_

Round 1

Reviewer 1 Report

The publication 'One-part alkali-activated materials: state of the art and perspectives' comprehensively demonstrates the applicability of just add water activated geopolymers. The authors critically present the opportunities and risks by comparing the above-mentioned method of producing geopolymers with the conventional method. 

The references presented comprehensively provide an overview of the current state of knowledge. The title corresponds to the content. 

The English included in the publication is correct.

I recommend the paper for publication.

Author Response

Reviewer #1:

The publication 'One-part alkali-activated materials: state of the art and perspectives' comprehensively demonstrates the applicability of just add water activated geopolymers. The authors critically present the opportunities and risks by comparing the above-mentioned method of producing geopolymers with the conventional method.

The references presented comprehensively provide an overview of the current state of knowledge. The title corresponds to the content.

The English included in the publication is correct.

I recommend the paper for publication.

Re:The authors thank you for your support of the study, and the full text has been reviewed in the revised manuscript.

Reviewer 2 Report

Overall, the paper is well written and provides a critical review of the current advances in the area of one-part AAM, which is quite interesting to the readers of the journal. Here are some suggestions for improvement:

- line 55-56: please add the biogenic acid resistance as well. check https://doi.org/10.1016/j.conbuildmat.2022.127912

- line 96-97: It should be 96 noted that the carbon emissions of the AAM are much lower than OPC .... This statement is a bit exaggerated since the environmentally friendly performance of AAM is still an area of debate among researchers. Please refer to https://doi.org/10.1016/j.jclepro.2011.03.012 and https://doi.org/10.1533/9781782422884.5.663

- line 116-117the small amount of gypsum .... this seems irrelevant to this section please remove it or explain the reason for toughness in the discussion.

- ref [60] in Table 1does not match the reference list. please revise.

- please add ref [91] to Table 1

- what about the current state of the art on the rheology of one-part AAM? please check 10.3151/jact.20.139 and 10.1016/j.cemconcomp.2021.104061

- line 193: Please add the use of fluorescent lamp waste glass powder as an activator https://doi.org/10.1016/j.conbuildmat.2022.129020

-section 2.6: please add the following paper to your review 10.18702/acf.2019.12.5.2.37

Author Response

Reviewer #2:

Overall, the paper is well written and provides a critical review of the current advances in the area of one-part AAM, which is quite interesting to the readers of the journal. Here are some suggestions for improvement:

The authors thank the reviewer for the valuable and careful comments. All of the following comments have been addressed in the revised version of the manuscript.

  1. Line 55-56: please add the biogenicacid resistance as well. Check  https://doi.org/10.1016/j.conbuildmat.2022.127912.

The authors thank the reviewer for your citation suggestions and literature on biogenic acid resistance has been added to the revised manuscript (line 56, ref [19]).

  1. Line 96-97: It should be 96 noted that the carbon emissions of the AAM are much lower than OPC .... This statement is a bit exaggerated since the environmentally friendly performance of AAM is still an area of debate among researchers. Please refer to https://doi.org/10.1016/j.jclepro.2011.03.012 and https://doi.org/10.1533/9781782422884.5.663.

Thank you very much for your more accurate point. According to the literature reports, more rigorous language has been used to avoid exaggerating situation in the revised manuscript, such as “…show that a part of geopolymer concrete has similar or even higher environmental impacts than OPC” (lines 96-99).

  1. Line 116-117 the small amount of gypsum .... this seems irrelevant to this section please remove it or explain the reason for toughness in the discussion.

Agree. The authors thank the editor for the careful review. It has been removed the description of gypsum in the revised manuscript (line 129).

  1. Ref [60] in Table 1 does not match the reference list. please revise.

The authors thank the reviewer for the careful review. The incorrect reference has been corrected, the correct reference number is ref [68].

  1. Please add ref [91] to Table 1

Thanks for your proposal. The references have been added to Table 1 and reordered in the revised manuscript.

  1. What about the current state of the art on the rheology of one-part AAM? please check 10.3151/jact.20.139 and 10.1016/j.cemconcomp.2021.104061.

The authors thank you for your advice. At present, rheological studies on OP-AAM are still slightly lacking. According to the literature provided by the reviewer and the latest reports, the current state of the art on rheology is added in Section 3.2 (lines 434-458).

  1. Line 193: Please add the use of fluorescent lamp waste glass powder as an activator https://doi.org/10.1016/j.conbuildmat.2022.129020

This is a very novel alkali source material, and this suggestion is very meaningful. The revision has been made in the revised manuscript (line 216).

  1. Section 2.6: please add the following paper to your review 10.18702/acf.2019.12.5.2.37

Thank you very much for your supplement to the reference. In line 354, we added the fiber-improving effect on elastic modulus and added the use of PE fiber in Table3 (lines 357-360).

Reviewer 3 Report

1.     Typically, chemical symbols and abbreviation are defined in the first instance and then used exclusively throughout the rest of the manuscript (i.e., the full name does not need to write out again). Please ensure that the chemical formula/symbols and all other abbreviations are defined and used in this manner

2.     Line 20. This word in abstract “maintenance regimes” is not displayed in the whole text. Please check this word.

3.     Line 121-123. Please further elaborate how these precursors materials affect the mechanical properties of the OP-AAM. Figure 4 has what relation with the mechanical properties? Figure 4 display the ternary diagram of the element composition after the hydraulic reaction and pozzolanic reaction? Or the element composition of the raw materials? Please clarify.

4.     Line 295. Please explain “risk of weathering”.

5.     Table 3. The star-shape icon indication is not defined. It is better to clearly state how and what the properties (workability/ strength/ chemical stability) are compared to, or what benchmark did you use to determine these performances. Reference 35, 50 and 94 did not discuss on the fluidity, setting time or slump of the OP-AAM influenced by the fibers. Kindly clarify how do you define “workability”. What does “strength” represent? The strength of the fiber or the strength of the fiber-reinforced OP-AAM? Overall, I think this table should be restructured.

6.     Line 337–339. What do you mean by “loss of composition during curing process”?

7.     Line 359-361. How does heat generation cause leaching the alkali components, then reduce the efficiency of polymer generation. Please further clarify

8.     Table 4. The content in the “evaluation” column could be combined with “advantage” and “shortage”.

9.     Section 3.2. Mechanical properties include compressive and flexural strength. it is suggested to add discussion on flexural strength of the OP-AAM, if any, or change the subtitle.

10.  Line 409-411. “The performance of other alkali sources, such as carbonates [39], aluminates [59,82], calcium oxides [20,66], and potassium oxides [78,107] is yet to be investigated.” Kindly check this statement.

In short, the paper focused on the most recent research on OP-AAM. But the lack of in-depth discussion regarding the chemical or physical interaction between the raw materials reduced the scientific value of this manuscript. For example, in line 323-324, “The best overall performance was at 1 vol% replacement of steel fiber, and the fiber length has a positive benefit to the improvement of bearing capacity.”, what was the bearing that you were referring and how does it contribute to the positive outcome? In line 374-375, the author could extend the discussion by explaining why the “setting time of the AAM is further reduced as the alkali content increases”, what was the reaction occurred that shorten the setting time? The critical thinking value was also less demonstrated. Polymers is a high-ranked journal. Hence I suggest a major correction for this paper.

Author Response

Reviewer #3:

In short, the paper focused on the most recent research on OP-AAM. But the lack of in-depth discussion regarding the chemical or physical interaction between the raw materials reduced the scientific value of this manuscript. For example, in line 323-324, “The best overall performance was at 1 vol% replacement of steel fiber, and the fiber length has a positive benefit to the improvement of bearing capacity.”, what was the bearing that you were referring and how does it contribute to the positive outcome? In line 374-375, the author could extend the discussion by explaining why the “setting time of the AAM is further reduced as the alkali content increases”, what was the reaction occurred that shorten the setting time? The critical thinking value was also less demonstrated. Polymers is a high-ranked journal. Hence I suggest a major correction for this paper.

The authors thank the reviewer for the valuable and careful comments. All of the following comments have been addressed in the revised version of the manuscript. At a suitable volume content, the fiber can show a positive behavior of compressive strength, while exceeding a certain threshold may lead to the weakening of mechanical properties due to the generation of a weak interfacial transition zone (lines 347-350). In lines 346-347, “The best overall performance was at 1 vol% replacement of steel fiber, and the fiber length has a positive benefit to the improvement of load-carrying capacity.” This view of point can be confirmed by the research of Shah et al [114].

Besides, The gels of polymeric products depend on the dissolution of precursors. In a highly alkaline environment, Si and Al are dissolved and polymerized to form hydraulic binders. Increasing the alkali content will accelerate this process. Finally, the author further reviewed the literature and added more mechanism explanations for readers to increase readability and ease of understanding (lines 96-99, lines 111-118, lines 131-135, lines 388-390, lines 407-410, etc.).

ADDITIONAL REFERENCES

  1. Ali Shah, S.F.; Chen, B.; Oderji, S.Y.; Haque, M.A.; Ahmad, M.R. Comparative Study on the Effect of Fiber Type and Content on the Performance of One-Part Alkali-Activated Mortar. Construction and Building Materials 2020, 243, 118221, doi:10.1016/j.conbuildmat.2020.118221.
  2. Typically, chemical symbols and abbreviation are defined in the first instance and then used exclusively throughout the rest of the manuscript (i.e., the full name does not need to write out again). Please ensure that the chemical formula/symbols and all other abbreviations are defined and used in this manner.

Thanks. In revised manuscript, the authors have carefully checked again. 

  1. Line 20. This word in abstract “maintenance regimes” is not displayed in the whole text. Please check this word.

The authors thank the reviewer for your careful review. The maintenance regimes were originally described as curing methods, so it has been changed to "curing methods" in the revised manuscript (line 20).

  1. Line 121-123. Please further elaborate how these precursors materials affect the mechanical properties of the OP-AAM. Figure 4 has what relation with the mechanical properties? Figure 4 display the ternary diagram of the element composition after the hydraulic reaction and pozzolanic reaction? Or the element composition of the raw materials? Please clarify.

Thanks for your proposal. The effect of these precursors on the mechanical properties of OP-AAM has been introduced in the revised manuscript. Since GGBS, FA, and MK are the most widely used precursors, a description of their properties has been added to the literature review. Fig.4 is used to comprehensively summarize the chemical composition of different precursor materials. According to the calcium content, they can be divided into two types of reactions. According to the  Ca/Si and Si/Al ratio, hydration reaction have different net structures. Precursors with high calcium content are more likely to form C-A-S-H gels. Such hydration products have higher mechanical properties, but there are risks in shrinkage and corrosion resistance. However, N-A-S-H gel formed by low calcium material has a slightly lower compressive strength than the high calcium, but it has excellent corrosion resistance and low shrinkage behavior. Different hydration products lead to various mechanical properties. The mixed system may combine the advantages both of them, depending on the concentration of Ca2+, the hydration products may transform between C-A-S-H gels and N-A-S-H (lines 131-136 and 139-147). 

  1. Line 295. Please explain “risk of weathering”.

Thanks. Weathering refers to the formation of fine particles of parent rock by the wind. The word “weathering” has been changed to “efflorescence” in the revised manuscript, due to the confusion of the two meanings. Luukkonen et al mentioned in their study that too high alkali concentration causes efflorescence. This means alkali components may leach out, causing damage to the structure. So according to the reference, the revised statement should be "efflorescence" (line 318).

  1. Table 3. The star-shape icon indication is not defined. It is better to clearly state how and what the properties (workability/ strength/ chemical stability) are compared to, or what benchmark did you use to determine these performances. Reference 35, 50 and 94 did not discuss on the fluidity, setting time or slump of the OP-AAM influenced by the fibers. Kindly clarify how do you define “workability”. What does “strength” represent? The strength of the fiber or the strength of the fiber-reinforced OP-AAM? Overall, I think this table should be restructured.

Agree with your proposal. To further clarify the effect of fiber type on the performance of OP-AAM, Table3 has been restructured in the revised manuscript. The ambiguous "workability" and "strength" columns were removed. Thus, the dosage of fiber and corresponding mechanical properties were added. Through the comparison of these parameters, it can be a more clear statement (lines 357-360).

Table 3. Fiber type on the performance of One-part AAM

References

Fiber type

Dosage of fiber (Vol.)

Compressive strength

Flexural strength

Effect evaluation

Abdollahnejad et al. [113]

ST

PP

PVA

0.5% and 1.0%

ST: 53-64 MPa

PP: 45-60 MPa

PVA: 47-61 MPa

n.r.

ST can get the best mechanical properties and has the highest chemical stability, while PP/PVA may have chemical adhesion.

Abdollahnejad et al. [108]

PP

PVA

BA

1.5%

PP: 50 MPa

PVA: 56 MPa

BA: 52 MPa

Hybrid: 42-61 MPa

PP: 7.8 MPa

PVA: 10.7 MPa

BA: 11.3 MPa

Hybrid: 7.8-10.9 MPa

All groups showed a decreased compressive strength from the original state, PVA has a better mechanical performance.

Shah et al. [114]

ST

PVA

BA

0.5%, 1.0%, 1.5% and 2.0%

ST: 28-33 MPa

PVA: 25-30 MPa

BA: 29-31 MPa

ST: 2.8-3.1 MPa

PVA: 2.8-3.2 MPa

BA: 2.3-2.7 MPa

The strength decreased when dosage of fiber was beyond 1.5%, BA have bad adhesion with OP-AAM.

Abdollahnejad et al. [43]

ST

PVA

BA

CEL

1.0%

ST: 82 MPa

PVA: 76 MPa

BA: 33 MPa

CEL: 53 MPa

Hybrid: 54-72 MPa

ST: 10.7 MPa

PVA: 9.6 MPa

BA: 6.1 MPa

CEL: 5.2 MPa

Hybrid: 6-11 MPa

Except for ST, all single and hybrid fibers increased porosity and water absorption, ST can improve both fire resistance and freezing resistance.

Perumal et al. [62]

ST

GLA

BA

1.0%

ST: 150-233 MPa

GLA: 143-177 MPa

BA: 135-168 MPa

n.r.

ST outperformed other fibers in AAM, and longer the fiber the better was the load carrying capacity. Mineral fibers were unstable in high alkaline environment.

Alrefaei and Dai [115]

ST

PE

0.5%, 1.0%, 1.5% and 2.0%

ST: 60-81 MPa

PE: 45-63 MPa

ST: 6.6-8.2 MPa

PE: 7.3-8.0 MPa

PE showed a comparative modulus of rupture relative to that of ST, and ST showed minor effects on flexural cracking and ultimate strengths.

Notes: ST means steel fiber; PP means polypropylene fiber; PVA means polyvinyl alcohol fiber; BA means basalt fiber; CEL means cellulose; GLA means glass fiber; PE means polyethylene fiber; n.r. indicates data not reported.

ADDITIONAL REFERENCES

  1. Abdollahnejad, Z.; Mastali, M.; Falah, M.; Shaad, K.M.; Luukkonen, T.; Illikainen, M. Durability of the Reinforced One-Part Alkali-Activated Slag Mortars with Different Fibers. Waste Biomass Valor 2021, 12, 487–501, doi:10.1007/s12649-020-00958-x.
  2. 6 Perumal, P.; Nguyen, H.; Carvelli, V.; Kinnunen, P.; Illikainen, M. High Strength Fiber Reinforced One-Part Alkali Activated Slag Composites from Industrial Side Streams. Construction and Building Materials 2022, 319, 126124, doi:10.1016/j.conbuildmat.2021.126124.
  3. 1 Abdollahnejad, Z.; Mastali, M.; Woof, B.; Illikainen, M. High Strength Fiber Reinforced One-Part Alkali Activated Slag/Fly Ash Binders with Ceramic Aggregates: Microscopic Analysis, Mechanical Properties, Drying Shrinkage, and Freeze-Thaw Resistance. Construction and Building Materials 2020, 241, 118129, doi:10.1016/j.conbuildmat.2020.118129.
  4. 1 Abdollahnejad, Z.; Mastali, M.; Luukkonen, T.; Kinnunen, P.; Illikainen, M. Fiber-Reinforced One-Part Alka-li-Activated Slag/Ceramic Binders. Ceramics International 2018, 44, 8963–8976, doi:10.1016/j.ceramint.2018.02.097.
  5. 1 Ali Shah, S.F.; Chen, B.; Oderji, S.Y.; Haque, M.A.; Ahmad, M.R. Comparative Study on the Effect of Fiber Type and Content on the Performance of One-Part Alkali-Activated Mortar. Construction and Building Materials 2020, 243, 118221, doi:10.1016/j.conbuildmat.2020.118221.
  6. 1Alrefaei, Y.; Dai, J.-G. Deflection Hardening Behavior and Elastic Modulus of One-Part Hybrid Fiber-Reinforced Geopolymer Composites. acf 2019, 5, 37–51, doi:10.18702/acf.2019.12.5.2.37.
  7. Line 337–339. What do you mean by “loss of composition during curing process”?

Thanks for your proposal. The phrase “loss of composition” means alkaline component leaching out, which leads to a decrease in alkali concentration. Generally, due to the concentration gradient being different between matrix and pore solution, that may result in alkaline components leaching out or water evaporation in thermal curing. Thus, the decrease in alkali concentration could be bad for the hydration reaction. The revision has been made in the manuscript (line 366).

  1. Line 359-361. How does heat generation cause leaching the alkali components, then reduce the efficiency of polymer generation. Please further clarify.

Thanks for your proposal. There is still some controversy about the effect of curing methods on AAM. However, some studies have shown that water curing is not conducive to the initial hydration reaction after paste mixing and immediate immersion in water will dilute the activator, thus reducing the efficiency of polymer reaction. The above statement has been changed in the revised manuscript (line 388-390).

  1. Table 4. The content in the “evaluation” column could be combined with “advantage” and “shortage”.

The authors thank you for your proposed improvements. Table 4 has been restructured in the revised manuscript. (lines 394-397)

Table 4 Comparison of different curing methods on one-part AAM

References

Curing method

Mix parameter

Major results

Evaluation

Abdollahnejad et al.[113]

Sealing/water curing

GGBS/ceramic + Na2SiO3

Water curing strength +3%~+13%

Water curing has better properties to compressive strength and flexural strength

Abdollahnejad et al. [118]

Thermal curing

GGBS/ceramic (porcelain

and raw) + Na2SiO3

Compressive strength -20%~+25%

Thermal curing led to the maximum improvement of strength

Ahmad et al. [92]

Ambient/water/thermal curing

FA/GGBS + Na2SiO3

Water curing strength -92~-60%; thermal curing strength +5~+135%

Water curing had an adverse effect

Haruna et al. [89]

Ambient/solar/water curing

Class C FA + Na2SiO3

Water curing strength -3%~-7%;

Solar curing strength +6%~+23%

Solar curing had the best compressive strength; ambient curing had nice flexural strength

Alzaza et al. [81]

Subzero curing (-5/-10/-20℃)

GGBS + Na2SiO3

Subzero curing strength -45%~-86%

The reactivity of AAS at low temperature is better than that of OPC, the strength of AAS and OPC is 68MPa, 9.7MPa, respectively.

Alzaza et al. [55]

Subzero curing (-5/-10/-20℃)

GGBS/SMP + Na2SiO3

Compressive strength under -5℃ is much higher then -10 and -20℃

Low temperature hinders the reaction process, and add SMP can get a better performance under subzero curing

Wei et al. [73]

Ambient/water curing

FASB + Na2SiO3/Na2CO3

Water curing strength -14%~-20%

Water curing decreases the quantity of hydration products and makes the microstructure much coarser

Notes: PCW means porcelain ceramic waste; GGBS means ground granulated blast-furnace slag; FA means fly ash; NS means nickel slag; FA(C) means C-class fly ash; SMP means submicron particle; FASB means fly ash sinking beads

ADDITIONAL REFERENCES

  1. Alzaza, A.; Ohenoja, K.; Illikainen, M. Enhancing the Mechanical and Durability Properties of Subzero-Cured One-Part Alkali-Activated Blast Furnace Slag Mortar by Using Submicron Metallurgical Residue as an Additive. Cement & Concrete Composites 2021, 122, 104128, doi:10.1016/j.cemconcomp.2021.104128.
  2. 7 Wei, T.; Zhao, H.; Ma, C. A Comparison of Water Curing and Standard Curing on One-Part Alkali-Activated Fly Ash Sinking Beads and Slag: Properties, Microstructure and Mechanisms. Construction and Building Materials 2021, 273, 121715, doi:10.1016/j.conbuildmat.2020.121715.
  3. Alzaza, A.; Ohenoja, K.; Illikainen, M. One-Part Alkali-Activated Blast Furnace Slag for Sustainable Construction at Subzero Temperatures. Construction and Building Materials 2021, 276, 122026, doi:10.1016/j.conbuildmat.2020.122026.
  4. Haruna, S.; Mohammed, B.S.; Wahab, M.M.A.; Liew, M.S. Effect of Paste Aggregate Ratio and Curing Methods on the Performance of One-Part Alkali-Activated Concrete. Construction and Building Materials 2020, 261, 120024, doi:10.1016/j.conbuildmat.2020.120024.
  5. 9 Ahmad, M.R.; Chen, B.; Shah, S.F.A. Influence of Different Admixtures on the Mechanical and Durability Properties of One-Part Alkali-Activated Mortars. Construction and Building Materials 2020, 265, 120320, doi:https://doi.org/10.1016/j.conbuildmat.2020.120320.
  6. 1Abdollahnejad, Z.; Mastali, M.; Luukkonen, T.; Kinnunen, P.; Illikainen, M. Fiber-Reinforced One-Part Alkali-Activated Slag/Ceramic Binders. Ceramics International 2018, 44, 8963–8976, doi:10.1016/j.ceramint.2018.02.097.
  7. 1 Abdollahnejad, Z.; Luukkonen, T.; Mastali, M.; Kinnunen, P.; Illikainen, M. Development of One-Part Alkali-Activated Ceramic/Slag Binders Containing Recycled Ceramic Aggregates. J. Mater. Civ. Eng. 2019, 31, 04018386, doi:10.1061/(ASCE)MT.1943-5533.0002608.
  8. Section 3.2. Mechanical properties include compressive and flexural strength. it is suggested to add discussion on flexural strength of the OP-AAM, if any, or change the subtitle.

The authors thank the reviewer for your comments. This is a rigorous point. In revised Section 3.3, the mechanical properties discussed include compressive strength only, since it is the most common evaluation index. So we took your suggestion seriously and changed the subtitle to "Compressive strengths" to fit the discussion in this section (line 459). 

  1. Line 409-411. “The performance of other alkali sources, such as carbonates [39], aluminates [59,82], calcium oxides [20,66], and potassium oxides [78,107] is yet to be investigated.” Kindly check this statement.

We have carefully checked this statement and discreetly review the studies. Now the revision has been made (line 471).

Round 2

Reviewer 3 Report

The author addressed every comment given. However, I still have 2 comments regarding the revised manuscript.

1.     Lines 135-136: “…but MK obtains a lower shrinkage rate and suitable setting time [52].”

The setting time of the MK-based OP AAM is suitable for what purpose/application? Please be concise. Or you may consider: “… but MK obtains a lower shrinkage rate and shorter/longer setting time compared to the FA-based OP AAM.”, if you are making comparison.

2.     There are obvious spelling and grammar errors, especially in section 2.1.

a.      The two reaction modes have different effects on the performance of AAM, which the hydraulic reaction is more likely to curing at ambient temperature. (lines 142-144). The second clause is confusing.

b.     Furthermore, it has… (line 144). What does the “it” refers to?

c.      …more suitable for use in environments containing erosive salts [58]. (line 147). Could change to : “…more suitable for environments that contain erosive salts [58].

It is crucial that you proofread the paper carefully again.

Author Response

The author addressed every comment given. However, I still have 2 comments regarding the revised manuscript.

The authors thank the reviewer for the valuable and careful comments. All of the following comments have been addressed in the revised version of the manuscript.

  1. Lines 135-136: “…but MK obtains a lower shrinkage rate and suitable setting time [52].”The setting time of the MK-based OP AAM is suitable for what purpose/application? Please be concise. Or you may consider: “… but MK obtains a lower shrinkage rate and shorter/longer setting time compared to the FA-based OP AAM.”, if you are making comparison.

Thanks. The MK is for comparison with GGBS, so the corresponding modification has been changed in the revised manuscript. 

  1. There are obvious spelling and grammar errors, especially in section 2.1.
  2. The two reaction modes have different effects on the performance of AAM, which the hydraulic reaction is more likely to curing at ambient temperature. (lines 142-144). The second clause is confusing.
  3. Furthermore, it has… (line 144). What does the “it” refers to?
  4. …more suitable for use in environments containing erosive salts [58]. (line 147). Could change to : “…more suitable for environments that contain erosive salts [58].

It is crucial that you proofread the paper carefully again..

The authors thank the reviewer for your careful review. In revised manuscript, the authors have carefully checked again. For questions a and b, the hydraulic reaction is more likely to curing at ambient temperature, it might cause high shrinkage and potential carbonization risk. So it refers to hydraulic reaction. The second clause has no explanatory relation with the main clause. Break them into two separate sentences for better understand (lines 143-145). In the revised manuscript, the sentence has been modified as "more suitable for environments that contain erosive salts".

According to reviewers common, the manuscript has been proofread carefully.
